# Hierarchically Metric-Structured Knowledge Graph Embeddings

## Abstract

In the vast landscape of big data, there is an important challenge in understanding data and structuring it in a suitable format. Knowledge graphs are considered a sophisticated solution to organize and infer data and knowledge, offering a structured framework that transcends disciplinary boundaries in medicine, culture, biology, social networks, music, and beyond. Despite their informativeness, these systems are typically incomplete and their intrinsic structure unknown, whereas existing methodologies for predicting missing facts and characterizing their structure face scalability and interpretability issues. Addressing this gap, we introduce a new latent feature model, leveraging the prominent RESCAL framework to account for degree heterogeneity, multiscale structure, and scalable inference using an approximation of the full likelihood of all triplets circumventing negative sampling inference strategies. This not only enhances computational efficiency but also provides deeper insights into the intrinsic multiscale structure of knowledge graphs, thereby advancing the interpretability of predictive models and paving the way for a more comprehensive understanding of complex information networks.

## 1 Introduction

Graphs, as a fundamental data structure, find application across diverse domains, including music, medicine, social networks, and more (Newman, 2003). The versatility of graphs is reflected in various structural manifestations such as unipartite graphs, bipartite graphs, and higher-order graphs such as knowledge graphs. A knowledge graph serves as a structured repository of information, encapsulating entities, relations, and semantic descriptions. Typically characterized by triples, these graphs represent factual relationships between two entities, thereby offering a comprehensive understanding of interconnected data. Some of the well-known curated knowledge graphs are YAGO (Suchanek et al., 2007), DBpedia (Auer et al., 2007), NELL (Carlson et al., 2010), Freebase (Bollacker et al., 2008), and Google Knowledge Graph (Google, May 2012). Despite the richness of information encapsulated within knowledge graphs, their inherent complexity surpasses conventional networks. This complexity necessitates the development of algorithms capable of mitigating modeling intricacies, especially when considering large-scale applications.

An often encountered challenge with knowledge graphs is their tendency to be incomplete and sparse, often lacking accurate factual information. A primary focus lies in the task of link prediction or knowledge graph completion, crucial for inferring missing connections within graphs and thereby completing the knowledge representation (Ali et al., 2021). The many research efforts developing and evaluating link prediction algorithms, particularly in real-world scenarios marked by sparsity, have underscored the potential of such methods to enhance the overall quality and utility of knowledge graphs significantly.

One of the important issues with knowledge graph completion is that knowledge graphs usually have only positive samples (i.e., triplets). Conventional methods often adopt a closed-world assumption, treating all non-positive triplets as negative, yet this may lead to scalability issues due to the potentially large number of false facts. Alternatively, some approaches use known constraints on the structure of a knowledge graph to generate more informative negative samples. However, how to devise the negative sampling procedure is non-trivial and can influence the learned representations, see also Nickel et al. (2015); Mishra et al. (2023)

for a discussion. In our approach, we mitigate scalability concerns by integrating a hierarchical multi-scale approximation into the training process (Nakis et al., 2023) within the closed-world assumption framework. This strategy effectively provides an accurate estimation of the full likelihood of our model and thereby substantially reduces the computational overhead associated with evaluating the full likelihood for accurate learning while circumventing the need for negative sampling strategies.

Knowledge graphs are structured data that can be described further with semantic information (Stamou & Chortaras, 2017). Previous studies have utilized techniques such as latent class modeling approaches partitioning entities into concepts (Kemp et al., 2006) as well as inferring concept hierarchies through hierarchical clustering approaches to organize entities structured using the learned hierarchical organization (Roy et al., 2006; Nickel & Tresp, 2011; Pietrasik & Reformat, 2020), thereby offering a richer and more descriptive arrangement of data rooted in taxonomies. Our approach inherently provides a similar structure because of the hierarchical multi-scale approximation used in the training process. Thus, we naturally obtain a taxonomy of the entities within a knowledge graph as part of the inference. Importantly, our approach thereby naturally bridges conventional knowledge graph embedding approaches with latent class and hierarchical modeling procedures by invoking a multi-scale hierarchical clustering structure designed to optimally characterize the knowledge graph by approximating the full data likelihood.

In our investigation, we prioritize structural models due to their versatility in application to various knowledge graphs, eliminating the necessity for contextual information. To learn latent representations of relational data, we focus on RESCAL (Nickel et al., 2011; Krompaß et al., 2013), a method that leverages a tensor factorization model designed to consider the inherent structure of relational data and has recently been confirmed as a current state-of-the-art structural model (Ruffinelli et al., 2020; Kong et al., 2019). We demonstrate how the RESCAL model can be reparameterized, transforming it into a distance model with random effects providing a seamless integration of hierarchical clustering into the training process while at the same time enabling to explicitly account for relation-specific degree heterogeneity. In this manner, we provide an accurate full-likelihood approximation, leveraging Nakis et al. (2023), importantly circumventing issues of negative sampling in conventional knowledge graph learning. This novel approach thereby allows us to capture the structural intricacies of a knowledge graph effectively. By defining latent representations for relation-specific nodes within the graph, we unveil compelling hierarchical clustering patterns, thereby enhancing our understanding of the underlying data in terms of the learned hierarchical (i.e., taxonomic) representation. Specifically, our contributions are:

- We demonstrate how RESCAL can be reformulated as a latent distance model with random effects.

- We thereby address the challenge of negative sampling and at the same time bridge knowledge graph embedding approaches with concept hierarchy learning providing a scalable accurate hierarchical full likelihood approximation by exploring the metric properties of our reformulation.

- On a variety of benchmark knowledge graphs, we show how our proposed model enables direct inference of concept hierarchies, while also providing reasonable link prediction when the embedding dimension is set sufficiently high.

The paper is organized as follows. In Section 2, we review related work on knowledge graph completion tasks. In Section 3, we present the RESCAL procedure and introduce the proposed Hierarchically Metric Structured Knowledge Graph Embeddings (HMSKGE) method. Section 4 details and discusses the results of our models on two tasks: link prediction and taxonomy induction. Finally, Section 5 concludes the paper by summarizing the findings, addressing limitations, and discussing the broader impact of the work.

## 2 Related Work

The field of Knowledge Graph Completion (KGC) comprises a wide range of methods (Wang et al., 2021), which can be broadly classified into three categories: structural, contextual, and hybrid approaches. Structural methods exploit the topology and relational structure of the knowledge graph, contextual methods incorporate external semantic or textual information, while hybrid approaches combine both sources. Within

structural methodologies, a further distinction can be made between latent representation learning approaches and methods based on explicit graph characteristics (Nickel et al., 2015). Latent representation approaches include factorization-based methods, neural representation learning models, and distance-based embeddings, whereas explicit graph-based approaches rely on symbolic reasoning, logical rules, or relational path exploration.

Among latent representation learning approaches, tensor factorization techniques have played a foundational role in KGC. Representative methods include Bayesian Clustered Tensor Factorization (Sutskever et al., 2009), TOPHITS (Kolda et al., 2005), PITF (Rendle & Schmidt-Thieme, 2010), RESCAL (Nickel et al., 2011; Krompaß et al., 2013), and TuckerER (Balažević et al., 2019). Matrix factorization methods have also been widely explored (Jiang et al., 2012; Riedel et al., 2013; Huang et al., 2014). Unlike tensor factorization approaches explicitly accounting for the mode structure of subject, object, and relations in knowledge graphs, matrix factorization techniques represent a knowledge graph in matrix form, where rows correspond to subject–object pairs and columns represent relations.

Neural latent representation models further extend these approaches by learning non-linear interactions between entities and relations. Representative examples include Multilayer Perceptron (MLP)-based architectures such as E-MLP and ER-MLP, as well as the Neural Tensor Network (NTN) (Socher et al., 2013; Dong et al., 2014). In contrast to classical factorization methods, these approaches leverage hidden layers to capture more expressive relational patterns.

Another important family of latent representation methods introduces distance-based modeling assumptions, where the likelihood of relationships is determined by distances or transformations in latent space. Representative models include Structured Embeddings (SE) (Bordes et al., 2011), TransE (Bordes et al., 2013), and RotatE (Sun et al., 2019).

In contrast, explicit graph-based approaches exploit observable structural properties of the knowledge graph, including logical rules and relational paths. Representative methods include ALEPH (Muggleton, 1995) and the Path Ranking Algorithm (Lao & Cohen, 2010). Hybrid approaches further combine latent and symbolic information, as seen in ARE (Nickel et al., 2014) and stacking-based methods (Wolpert, 1992). These paradigms and their evolution in knowledge graph embedding are comprehensively reviewed in Cao et al. (2024).

Moving to contextual methodologies, the advent of Graph Neural Networks (GNNs) has significantly advanced knowledge graph completion. Models such as the Relational Graph Convolutional Network (R-GCN) (Schlichtkrull et al., 2018), GraIL (Teru et al., 2020), and Decagon (Zitnik et al., 2018) demonstrate the effectiveness of message-passing architectures for relational reasoning. In addition, language model-based approaches such as KG-BERT (Yao et al., 2019) and SimKGC (Wang et al., 2022) leverage pretrained representations to incorporate textual semantics. More recently, multilingual and text-augmented frameworks (Song et al., 2024; Jiang et al., 2023) have further improved the contextual grounding of entity–relation representations.

Hierarchical structure has also been incorporated into KGC models to capture concept hierarchies and taxonomic organization beyond standard relational patterns (Roy et al., 2006; Nickel & Tresp, 2011; Pietrasik & Reformat, 2020). As such, HAKE embeds entities in polar coordinate space, where the radial component encodes hierarchical levels, improving performance on hierarchy-rich graphs (Zhang et al., 2020). Hierarchical representations have also been explored in downstream applications such as recommendation, where HAKG models user–item subgraphs with hierarchical attention to capture multi-hop semantics (Sha et al., 2021). More recently, KG-FIT incorporates hierarchical structure through an LLM-guided clustering and refinement framework that injects external semantic knowledge into KGE learning (Jiang et al., 2024). A major advantage of this approach is its ability to jointly capture global semantic information from LLMs and local structural information from the KG, leading to improved performance on link prediction tasks. However, since the hierarchy is automatically generated from LLM outputs, the resulting structure may be affected by hallucinations and semantic inaccuracies, which can introduce noise into the learned embeddings.

Building upon these directions, the proposed HMSKGE framework combines ideas from tensor factorization, latent distance modeling, and hierarchical representation learning within a unified probabilistic framework

for knowledge graph completion that bridges knowledge graph embedding approaches with hierarchical representation learning of concepts.

## 3 Methodology

We employed the Resource Description Framework (RDF) format from the semantic web to depict knowledge graphs, employing triples composed of (subject, predicate, and object) to denote relationships (Decker et al., 2000). Within this framework, the predicates denote connections between entities.

We conceptualized a knowledge graph as a tensor, denoted as $\mathcal{X} \in \{0,1\}^{N_e \times N_e \times N_r}$, where $N_e$ represents the number of entities, and $N_r$ signifies the number of relations involved. Every element, denoted as $\mathcal{X}_{ijk}$, within the tensor, holds a value of one to indicate the existence of a relationship. This signifies that there is a connection between the $i - th$ and the $j - th$ entity regarding the $k - th$ predicate. Conversely, the entry is set to zero if a relationship does not exist or is unknown.

### 3.1 RESCAL

To learn latent representations of relational data, we focus on RESCAL (Nickel et al., 2011; Krompaß et al., 2013) - a current state-of-the-art structural model (Ruffinelli et al., 2020; Kong et al., 2019). According to the RESCAL model, each relational slice $\mathcal{X}_k$ of the tensor $\mathcal{X}$ is approximated by:

$$\mathcal{X}_k \approx \mathcal{E}\mathcal{R}_k\mathcal{E}^\top, for \ k = 1, 2, \ldots, N_r. \tag{1}$$

As a result, $\mathcal{E}$ represents a $N_e \times D$ matrix containing the latent component representation of entities within the domain, and $\mathcal{R}_k$ is an asymmetric $D \times D$ matrix that characterizes the interactions of latent components associated with the $k - th$ predicate corresponding to a slice of the tensor $\mathcal{R} \in \mathbb{R}^{D \times D \times N_r}$.

In the original work on RESCAL model, (Nickel et al., 2011) estimation was based on least squares minimization. However, given the binary nature of tensor $\mathcal{X}$, we estimated Equation 1 by assuming that $\mathcal{X}_{ijk}$ follows a Bernoulli distribution which was demonstrated in (Nickel & Tresp, 2013) to enhance performance. This approach aligns with common practices in modeling network data, as highlighted in previous works (Hoff et al., 2002; Handcock et al., 2007; Krivitsky et al., 2009). Consequently, we define the BERNOULLI model and optimize $\mathcal{E}$ and $\mathcal{R}$ by employing stochastic gradient descent on the negative log-likelihood given by

$$\mathcal{L}(\mathcal{X}|\mathcal{E}, \mathcal{R}) := \sum_{i,j,k} x_{ijk}\eta_{ijk} - \log(1 + \exp(\eta_{ijk})), \quad \eta_{ijk} := \mathbf{e}_i \mathcal{R}_k \mathbf{e}_j^\top. \tag{2}$$

where $\mathbf{e}_i \in \mathbb{R}^D$ represent the embedding of node $i$.

### 3.2 The Hierarchically Metric Structured Knowledge Graph Embeddings

To enhance the model's versatility, we integrate random effects into the likelihood, thereby formulating the BERNOULLI-RE model. This extension is seamlessly achieved by adding $\beta_{ik}$ and $\gamma_{jk}$, in the definition of $\eta_{ijk}$ and enables direct modeling of degree heterogeneity across the predicates.

Prior studies (Wind & Mørup, 2012; Nakis et al., 2023) have shown that incorporating a Poisson likelihood given by

$$\mathcal{L}(\mathcal{X}|\mathcal{E}, \mathcal{R}, \beta, \gamma) := \sum_{i,j,k} x_{ijk} \log \lambda_{ijk} - \lambda_{ijk}, \quad \lambda_{ijk} := \exp\left(\mathbf{e}_i \mathcal{R}_k \mathbf{e}_j^\top + \beta_{ik} + \gamma_{jk}\right), \tag{3}$$

does not diminish a model's predictive accuracy and at the same time can be easily applied to weighted knowledge graphs. Additionally, Poisson models possess beneficial decoupling properties among predictor variables in the likelihood function (Karrer & Newman, 2011; Herlau et al., 2014; Nakis et al., 2023), which we utilize in the following to accurately approximate the full likelihood.

Our current modeling approach primarily utilizes pairwise latent interactions, where the likelihood of a triplet's realization is determined by evaluating $\mathbf{e}_i \mathcal{R}_k \mathbf{e}_j^\top$. Transitioning from the RESCAL model to a Latent Distance Model (LDM) (Hoff et al., 2002) is seamless when the model includes random effects and can be accomplished through reparameterization. In this modified model, the score assigned to each triplet is contingent on the distances between latent representations of entities and relations. Specifically, we defined the LDM model by setting $\lambda_{ijk}$ of Equation 3 as:

$$\lambda_{ijk} := \exp(\hat{\beta}_{ik} + \hat{\gamma}_{jk} - \frac{1}{2}\|\mathbf{e}_i \mathcal{V}_k - \mathbf{e}_j \mathcal{Q}_k\|_2^2) \tag{4}$$

where $\mathcal{V}_k$ and $\mathcal{Q}_k$ correspond to the $k^{th}$ slices of the tensors $\mathcal{V} \in \mathbb{R}^{D \times D \times N_r}$ and $\mathcal{Q} \in \mathbb{R}^{D \times D \times N_r}$ accounting for the relation-specific information such that $\mathcal{R}_k = \mathcal{V}_k \mathcal{Q}_k^\top$. Noting that

$$\|\mathbf{e}_i \mathcal{V}_k - \mathbf{e}_j \mathcal{Q}_k\|_2^2 = (\mathbf{e_i}\mathcal{V}_k - \mathbf{e}_j \mathcal{Q}_k)(\mathbf{e_i}\mathcal{V}_k - \mathbf{e}_j \mathcal{Q}_k)^\top = \mathbf{e_i}\mathcal{V}_k \mathcal{V}_k^\top \mathbf{e}_i^\top + \mathbf{e}_j \mathcal{Q}_k \mathcal{Q}_k^\top \mathbf{e}_j^\top - 2\mathbf{e}_i \mathcal{V}_k \mathcal{Q}_k^\top \mathbf{e}_j \top, \tag{5}$$

we obtain Equation 4 by defining $\beta_{ik} = \hat{\beta_{ik}} - \frac{1}{2}\mathbf{e}_i \mathcal{V}_k \mathcal{V}_k^\top \mathbf{e}_i^\top$ and $\gamma_{jk} = \hat{\gamma_{jk}} - \frac{1}{2}\mathbf{e}_j \mathcal{Q}_k \mathcal{Q}_k^\top \mathbf{e}_j^\top$.

Importantly, this reparameterization to a conventional LDM model (Equation 4) with random effects ($\beta_{ik}$ and $\gamma_{jk}$) enables us to explore the hierarchical block approximation proposed in the context of the LDM model in Nakis et al. (2023). Our goal is to streamline the overall likelihood computation while structuring our entities into coherent hierarchies or categories. This is accomplished by constructing a block-like hierarchical arrangement using a clustering approach applied to latent variables in Euclidean space. We organized the embedded clusters into a hierarchy using a tree structure, defining a cluster dendrogram forming the Hierarchically Metric Structured Knowledge Graph Embeddings (HMSKGE) model. The HMSKGE is defined by the following loss function:

$$\begin{aligned}
\mathcal{L}(\mathcal{X}|\mathcal{E}, \mathcal{V}, \mathcal{Q}, \beta, \gamma) &:= \sum_{x_{i,j,k}=1}\left(\beta_{ik} + \gamma_{jk} - \frac{1}{2}\|\mathbf{e}_i\mathcal{V}_k - \mathbf{e}_j\mathcal{Q}_k\|_2^2\right) - \sum_{i,j,k}\exp(\beta_{ik} + \gamma_{jk} - \frac{1}{2}\|\mathbf{e}_i\mathcal{V}_k - \mathbf{e}_j\mathcal{Q}_k\|_2^2) \\
&\approx \sum_{x_{i,j,k}=1}\left(\beta_{ik} + \gamma_{jk} - \frac{1}{2}\|\mathbf{e}_i\mathcal{V}_k - \mathbf{e}_j\mathcal{Q}_k\|_2^2\right) - \sum_{p=1}^{P_L}\sum_{i,j\in C_p^{(L)}}\sum_k \exp(\beta_{ik} + \gamma_{jk} - \frac{1}{2}\|\mathbf{e}_i\mathcal{V}_k - \mathbf{e}_j\mathcal{Q}_k\|_2^2) \\
&- \sum_{l=1}^{L}\sum_{p=1}^{P_l}\sum_{p'\neq p}^{P_l}\sum_k \left(\exp(-\frac{1}{2}\|\mu_{pk}^{(l)} - \mu_{p'k}^{(l)}\|_2^2)\left(\sum_{i\in C_p^{(l)}}\exp(\beta_{ik})\right)\left(\sum_{j\in C_{p'}^{(l)}}\exp(\gamma_{jk})\right)\right)
\end{aligned} \tag{6}$$

where $l \in \{1, \dots, L\}$ denotes the $l$th dendrogram level, $p \in \{1, \dots, P_l\}$ indexes the cluster $C_p^{(l)}$ at that level, and $\mu_{pk}^{(l)}$ is the corresponding centroid for the $k$th relation.

In particular, the first term in Equation 6 captures the exact contribution of observed links, while the second term accounts for non-links within clusters at the finest level of the hierarchy, where interactions are computed explicitly between node pairs in the same cluster. The third term provides a hierarchical approximation of non-links between nodes in different clusters across all levels, replacing pairwise interactions with centroid-based distances weighted by aggregated node-specific effects. This decomposition substantially reduces the computational complexity of the non-link term while preserving the underlying geometric structure of the latent space. An example of an application of our model is given in Figure 1.

### 3.2.1 Dendrogram construction

The process of dendrogram construction initiates by identifying clusters within its initial layer, typically with a count approximately equal to the $\ln N_e$. This initial clustering is facilitated through the utilization of the standard K-MEANS algorithm applied to all entities. Herein, the latent representation of each entity is defined as the concatenation of $\mathbf{e}_i\mathcal{V}_k$ and $\mathbf{e}_i\mathcal{Q}_k$ for all relations $k \in 1, \dots, K$, i.e. $\mathbf{z}_i = [\mathbf{e}_i\mathcal{V}_1 \ \mathbf{e}_i\mathcal{Q}_1, \dots, \mathbf{e}_i\mathcal{V}_K \ \mathbf{e}_K\mathcal{Q}_K]$

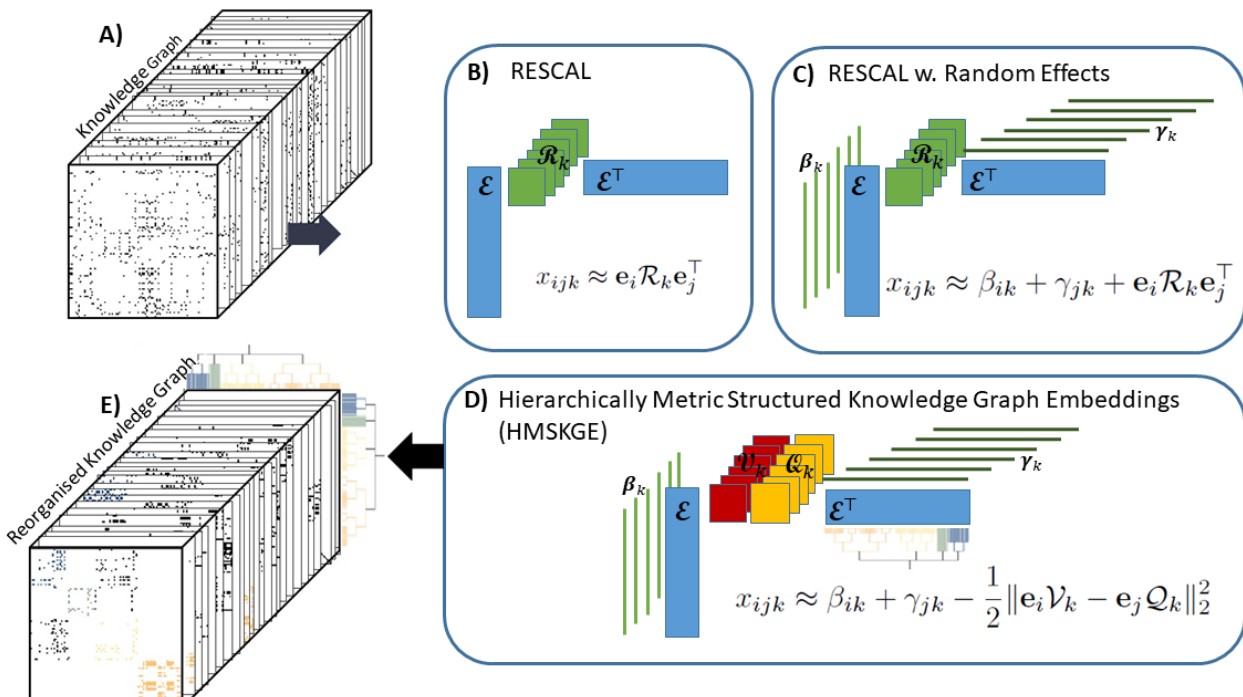

Figure 1: Illustration of the HMSKGE applied to the *Kinships* knowledge graph. **A)** The *Kinships* knowledge graph. **B)** The RESCAL model decomposes the knowledge graph into latent factors $\mathcal{E}$ as well as relation-specific interactions $\mathcal{R}_k$ originally solved using the least-squares loss but advanced to Bernoulli (cross-entropy) with improved results. **C)** To explicitly account for degree heterogeneity we include random effects and explore that random effects efficiently decouple in the inference by utilizing the Poisson loss that can also account for integer-weighted graphs. **D)** Exploring the RESCAL formulation with random effects we show that the model can be reformulated as a distance model admitting efficient and accurate full likelihood approximation not relying on negative sampling procedures during inference. **E)** Visualization results when reorganizing the nodes according to the inferred hierarchical structure learned by organizing the knowledge graph in coherent groups at multiple scales of the inferred hierarchy.

allowing for a comprehensive consideration of the position of all entity relations across the predicates during the clustering procedure performed on $\mathbf{Z} \in \mathbb{R}^{N_e \times N_r \cdot 2 \cdot D}$. Consequently, centroids per relation and cluster can be derived and represented as $\mu_p \in \mathbb{R}^{N_r \cdot 2 \cdot D}$. Subsequently, every cluster is further divided through the application of the K-MEANS algorithm, resulting in the creation of two smaller clusters, effectively doubling the total count to 2 clusters. This subdivision process iterates until each cluster reaches a minimum threshold of $\ln(N_e)$ entities.

Crucially, the model formulation given in Equation 4 suffers from redundancies in the scales of $\mathbf{V}_k$ and $\mathbf{Q}_k$ such that the one can become large countered by the other becoming low in values and vice versa with the random effects also absorbing these changes in scale. When performing clustering in the space $\mathbf{z}_i = [\mathbf{e}_i \mathcal{V}_1 \; \mathbf{e}_i \mathcal{Q}_1, \dots, \mathbf{e}_K \mathcal{V}_K \; \mathbf{e}_K \mathcal{Q}_K]$ this in turns results in low variance components for some parts of the concatenation making the clustering pay little attention to these parts when performing the segmentation. This motivates introducing a scale normalisation step before constructing $\mathbf{z}_i$ for clustering. To remedy this, we ensure by reparameterising the model that $\mathbf{V}_k$ and $\mathbf{Q}_k$ have same magnitude using the following transformations of the parameters that leaves Equation 4 invariant:

$$\tilde{\mathcal{V}}_k(d) = \frac{\sqrt{\|\mathcal{V}_k(d)\|_F \|\mathcal{Q}_k(d)\|_F}}{\|\mathcal{V}_k(d)\|_F} \mathcal{V}_k(d), \qquad \tilde{\mathcal{Q}}_k(d) = \frac{\sqrt{\|\mathcal{V}_k(d)\|_F \|\mathcal{Q}_k(d)\|_F}}{\|\mathcal{Q}_k(d)\|_F} \mathcal{Q}_k(d),$$

$$\tilde{\beta}_{ik} = \beta_{ik} - \frac{1}{2}(\mathbf{I} - \frac{\|\mathcal{Q}_k\|_F}{\|\mathcal{V}_k\|_F})\mathbf{e}_i \mathcal{V}_k \mathcal{V}_k^\top \mathbf{e}_i^\top, \qquad \tilde{\gamma}_{jk} = \gamma_{jk} - \frac{1}{2}(\mathbf{I} - \frac{\|\mathcal{V}_k\|_F}{\|\mathcal{Q}_k\|_F})\mathbf{e}_j \mathcal{Q}_k \mathcal{Q}_k^\top \mathbf{e}_j^\top. \tag{7}$$

Importantly, the factors $\frac{\sqrt{\|\mathcal{V}_k(d)\|_F \|\mathcal{Q}_k(d)\|_F}}{\|\mathcal{V}_k(d)\|_F}$ and $\frac{\sqrt{\|\mathcal{V}_k(d)\|_F \|\mathcal{Q}_k(d)\|_F}}{\|\mathcal{Q}_k(d)\|_F}$ renormalize the $d^{\text{th}}$ column of $\mathcal{V}_k$ and $\mathcal{Q}_k$ respectively such that the columns of the renormalized matrices $\tilde{\mathcal{V}}_k$ and $\tilde{\mathcal{Q}}_k$ have identical scale of $\|\tilde{\mathcal{V}}_k(d)\|_F = \|\tilde{\mathcal{Q}}_k(d)\|_F = \sqrt{\|\mathcal{V}_k(d)\|_F \|\mathcal{Q}_k(d)\|_F}$, whereas the random effects absorb the change in squared norm when reparameterizing using $\|\tilde{\mathcal{V}}_k(d)\|_F^2$ instead of $\|\mathcal{V}_k(d)\|_F^2$ and using $\|\tilde{\mathcal{Q}}_k(d)\|_F^2$ instead of $\|\mathcal{Q}_k(d)\|_F^2$. Using these normalizations it can be trivially shown that

$$\exp(\beta_{ik} + \gamma_{jk} - \frac{1}{2}\|\mathbf{e}_i \mathcal{V}_k - \mathbf{e}_j \mathcal{Q}_k\|_2^2) = \exp(\tilde{\beta}_{ik} + \tilde{\gamma}_{jk} - \frac{1}{2}\|\mathbf{e}_i \tilde{\mathcal{V}}_k - \mathbf{e}_j \tilde{\mathcal{Q}}_k\|_2^2). \tag{8}$$

This re-normalization is applied at every iteration of the HMSKGE inference procedure.

### 3.2.2 Complexity

The computational complexity associated with estimating Equation 6 becomes linearithmic in the number of entities $N_e$ when terminating at clusters of size $\ln(N_e)$, reducing the overall cost of evaluating the full likelihood from $\mathcal{O}(N_e^2 N_r)$ to $\mathcal{O}(N_e \log(N_e) N_r)$. Compared with classical tensor factorization approaches such as RESCAL_ALS (complexity $\mathcal{O}(N_r N_e D^2 + N_r D^3)$) and probabilistic formulations like POISSON, BERNOULLI, and LDM (complexity $\mathcal{O}((|\mathcal{E}| + C)D^2)$, where $C$ denotes the number of negative samples), the proposed HMSKGE achieves a substantially lower complexity of $\mathcal{O}(N_e \log(N_e) N_r D^2)$. This hierarchical approximation eliminates the reliance on negative sampling, which, while common in existing methods, introduces stochastic variability and may neglect important entity interactions. By organizing entities according to their latent-space proximity, HMSKGE concentrates computational effort on local neighborhoods where accurate predictions are most critical, while concurrently producing an interpretable hierarchical taxonomy that reflects entity similarity. This combination of efficiency, scalability, and interpretability makes HMSKGE particularly well-suited for large-scale knowledge graph embedding and inference. We note that other hierarchical approximations can be employed instead of our dendrogram construction based on k-means. As such agglomerative clustering based on the squared Euclidean distance using Ward's method or average linkage based on the Euclidean distance could be used. However, these procedures rely on agglomeration based on all pairwise distances providing less favorable scaling properties than the above dendrogram construction by k-means clustering.

### 3.2.3 Embedding Proximity

The likelihood of a triplet in a knowledge graph can be related to the distance between the embeddings of the subject and object entities. Intuitively, entities that are closer in the embedding space are more likely to be connected. Lemma 1 formalizes this intuition by relating embedding proximity to the probability of a triplet.

**Lemma 1.** *Let* $\{\mathcal{X}_{ijk}\}_{(i,j,k) \in N_e \times N_e \times N_r}$ *be the set of independent random variables indicating the presence of edges and following a Poisson distribution. Then, each triple* $(i, j, k)$ *satisfies*

$$-\log(\varepsilon p_{ijk}) + \beta_{ik} + \gamma_{jk} \geq \frac{1}{2}\|\mathbf{e}_i \mathcal{V}_k - \mathbf{e}_j \mathcal{Q}_k\|_2^2 \tag{9}$$

*where* $p_{ijk} = P(\mathcal{X}_{ijk} > \varepsilon)$ *and* $\varepsilon$ *denotes the weight of the interaction.*

*Proof.* Since we suppose that each triple is an independent and identically distributed variable, by Markov's inequality, we can write that

$$P(\mathcal{X}_{ijk} > \varepsilon) \leq \frac{\mathbb{E}[\mathcal{X}_{ijk}]}{\varepsilon} = \frac{\lambda_{ijk}}{\varepsilon} = \frac{\exp\left(\beta_{ik} + \gamma_{jk} - \frac{1}{2}\|\mathbf{e}_i \mathcal{V}_k - \mathbf{e}_j \mathcal{Q}_k\|_2^2\right)}{\varepsilon} \tag{10}$$

which implies that

$$-\log\left(\varepsilon p_{ijk}\right) + \beta_{ik} + \gamma_{jk} \geq \frac{1}{2}\|\mathbf{e}_i \mathcal{V}_k - \mathbf{e}_j \mathcal{Q}_k\|_2^2 \tag{11}$$

$\square$

## 4 Results & Discussion

We conducted a series of experiments to evaluate the expressiveness of the embeddings generated by the HMSKGE model. These experiments were carried out on a Tesla A100 PCIE 80 GB GPU, and focused on link prediction as well as taxonomy induction. We used a diverse range of datasets of small and large sizes and compared our results against multiple baseline models to ensure a thorough evaluation.

The datasets employed in this study encompass *Kinships* (Quinlan, 1990), the *WN18RR* dataset (Dettmers et al., 2018), the *FB15k-237* dataset (Toutanova & Chen, 2015), and the *YAGO3-10* dataset (Dettmers et al., 2018). The *Kinships* knowledge graph specifically delineates kinship relations within the Alwayarra tribe, as documented by (Kemp et al., 2006). The *WN18RR* dataset is derived from the *WN18* subset of WordNet, while the *FB15k-237* dataset is a modified version of the *FB15k* dataset. The *FB15k-237* dataset is specifically designed based on Freebase entity pairs, eliminating inverse relations to address concerns related to the impact of test triplets generated by inverting training set triplets. The *YAGO3-10* is a subset of the *YAGO3* dataset that combines the information from Wikipedia articles in multiple languages with WordNet, GeoNames, and other data sources. The above datasets have been widely used as benchmarks for the link prediction task. Table 1 provides the statistics of all the datasets that we used.

Table 1: Statistics of the datasets considered in this work.

| Dataset | Triples | Entities | Relations | Mean-degree | Median-degree |
|---------|---------|----------|-----------|-------------|---------------|
| *YAGO3-10* | 1,179,040 | 123,182 | 37 | 12.640 | 8 |
| *FB15k-237* | 310,116 | 14,541 | 237 | 34.194 | 21 |
| *WN18RR* | 93,003 | 40,943 | 11 | 3.727 | 3 |
| *Kinships* | 10,790 | 104 | 26 | 105 | 105 |

For the baseline model, we selected RESCAL (Nickel et al., 2011), implemented using alternating least squares optimization during training. To systematically evaluate the proposed HMSKGE, we contrasted its derivation steps against the RESCAL baseline using multiple formulations: the Bernoulli likelihood (BERNOULLI), the Bernoulli likelihood with random effects (BERNOULLI-RE), and the Poisson likelihood (POISSON). We further considered a Poisson-based reformulation as a distance model, which we denote LDM.

All baseline models were trained using batching and negative sampling (Dettmers et al., 2018). This setup enabled us to quantify the performance gap between the hierarchical approximation of the full likelihood and conventional inference with negative sampling, using both the LDM and POISSON formulations. Implementation details and code are available in the GitHub repository[1].

### 4.1 Link Prediction

For the task of link prediction, also referred to as knowledge graph completion, we present results using two evaluation metrics: one rank-based metrics, as proposed by Ali et al. (2021); Hoyt et al. (2022), and one classification-based metric. The rank-based metric used is HITS@10, which evaluates the performance of ranking systems by measuring the proportion of true entities that appear within the top 10 ranks, with higher values indicating better performance. The classification-based metric, AUC-ROC, summarize the model's ability to discriminate between positive and negative instances across all classification thresholds.

---

[1]Not provided due to the double-blind review policy; provided as supplementary material.

These metrics capture complementary aspects of link prediction performance. Whereas AUC-ROC assesses the model's overall discriminative ability across all possible thresholds, HITS@10 evaluates whether the correct entity is prioritized among the highest-ranked candidates. Consequently, models may achieve strong AUC-ROC scores while exhibiting lower HITS@10 values, and vice versa.

The standard train–test split provided with each benchmark dataset was used, consistent with prior work. For the ranking-based evaluation, the model was trained on the training set and evaluated on the test set by performing head and tail prediction, as is standard in knowledge graph completion. Specifically, for each positive triplet in the test set, we replaced the head (or tail) entity in turn with every possible entity in the graph and ranked all resulting candidate triplets according to their predicted scores. The true entity's rank was then recorded, and the HITS@10 metric was computed as the average proportion of cases where the correct entity appeared within the top 10 ranks, averaged over both head and tail prediction tasks. For the AUC-ROC evaluation, we randomly created negative test samples following a balanced sampling strategy: an equal number of negative triplets were generated by randomly replacing either the head or tail entity in the positive test triplets, ensuring that the resulting triplets did not appear in the training or test sets. The model then produced a score for each positive and negative triplet using the learned embeddings, and the AUC-ROC was computed based on these scores and their corresponding binary labels.

In Tables 2 and 3, we report the results for RESCAL, HMSKGE, and the corresponding ablation models used as baselines. All models were trained with embedding dimensions of 2, 3 and 16, using a learning rate of 0.001 over 100,000 epochs. Our models consistently outperform the traditionally trained RESCAL, which relies on alternating least squares optimization, confirming that gradient descent optimization based on Bernoulli likelihoods yields superior performance, consistent with prior findings (Nickel & Tresp, 2013). Incorporating random effects further improves performance, likely by better capturing nodes that act as hubs in the network and efficiently characterizing degree heterogeneity within each relational layer. The BERNOULLI and POISSON variants exhibit similar behavior, in line with previous work (Wind & Mørup, 2012), and the POISSON and LDM models also show closely aligned performance patterns. The HMSKGE model performs in general reasonably well in the link prediction task when the embedding dimension is large with some deterioration in performance when using low dimensional $D \leq 3$. We attribute this to the more restricted model capacity impacting the hierarchical approximation.

Table 2: Comparison of HITS@10 and AUC-ROC scores for models on the *Kinships* and *WN18RR* datasets averaged over 3 runs. The standard error of the mean for all cases is approximately 0.001.

| Model | Metric (D) | *Kinships* | | | *WN18RR* | | |
|---|---|---|---|---|---|---|---|
| | | 2 | 3 | 16 | 2 | 3 | 16 |
| RESCAL | HITS@10 | 0.100 | 0.721 | **0.967** | 0.013 | 0.024 | 0.279 |
| | AUC-ROC | 0.959 | 0.977 | **0.974** | 0.788 | 0.849 | 0.849 |
| BERNOULLI | HITS@10 | 0.500 | 0.701 | 0.955 | 0.029 | 0.039 | 0.404 |
| | AUC-ROC | 0.971 | 0.956 | 0.965 | 0.923 | 0.938 | 0.899 |
| BERNOULLI-RE | HITS@10 | 0.698 | 0.854 | 0.937 | **0.043** | 0.044 | 0.407 |
| | AUC-ROC | 0.958 | 0.945 | 0.955 | 0.940 | **0.944** | 0.891 |
| POISSON | HITS@10 | 0.828 | **0.889** | 0.940 | 0.043 | **0.046** | **0.429** |
| | AUC-ROC | 0.958 | 0.947 | 0.955 | **0.941** | **0.944** | 0.885 |
| LDM | HITS@10 | **0.877** | 0.833 | 0.946 | 0.038 | 0.042 | 0.386 |
| | AUC-ROC | 0.948 | 0.961 | 0.957 | 0.922 | 0.942 | 0.866 |
| HMSKGE | HITS@10 | 0.203 | 0.196 | 0.673 | 0.017 | 0.004 | 0.162 |
| | AUC-ROC | 0.916 | 0.913 | 0.918 | 0.784 | 0.765 | **0.920** |

Based on the results in Tables 2 and 3, where HMSKGE showed limited performance in low-dimensional settings, we further evaluated RESCAL, HMSKGE, and our ablation baselines under varying large embedding sizes and training parameters (Tables 4 and 5). All models were trained with embedding dimensions

Table 3: Comparison of HITS@10 and AUC-ROC scores for models on the *FB15k-237* and *YAGO3-10* datasets averaged over 3 runs. The standard error of the mean for all cases is approximately 0.001.

| Model | Metric (D) | *FB15k-237* 2 | 3 | 16 | *YAGO3-10* 2 | 3 | 16 |
|---|---|---|---|---|---|---|---|
| RESCAL | HITS@10 | 0.126 | 0.132 | 0.142 | 0.022 | 0.021 | 0.010 |
| | AUC-ROC | 0.932 | 0.954 | 0.967 | 0.908 | 0.910 | 0.915 |
| BERNOULLI | HITS@10 | 0.138 | 0.265 | 0.448 | 0.049 | 0.031 | **0.068** |
| | AUC-ROC | 0.993 | 0.996 | **0.999** | 0.980 | 0.901 | **0.997** |
| BERNOULLI-RE | HITS@10 | 0.133 | 0.348 | 0.448 | **0.058** | 0.058 | 0.067 |
| | AUC-ROC | 0.995 | **0.997** | **0.999** | **0.986** | 0.987 | **0.997** |
| POISSON | HITS@10 | **0.326** | **0.352** | **0.449** | **0.058** | 0.058 | 0.064 |
| | AUC-ROC | **0.996** | **0.997** | 0.998 | **0.986** | 0.986 | 0.995 |
| LDM | HITS@10 | 0.277 | 0.314 | 0.425 | 0.054 | **0.059** | 0.045 |
| | AUC-ROC | 0.986 | **0.997** | 0.996 | 0.981 | **0.990** | 0.965 |
| HMSKGE | HITS@10 | 0.112 | 0.110 | 0.226 | 0.014 | 0.008 | 0.032 |
| | AUC-ROC | 0.843 | 0.899 | 0.996 | 0.852 | 0.899 | 0.938 |

of 20, and 30, a learning rate of 0.01, and for 20,000 epochs-except for WN18RR, which required 100,000 epochs and a reduced learning rate of 0.001. We used a smaller learning rate for WN18RR because its complex relational structure made training unstable at 0.01, lowering it to 0.001 ensured smoother convergence during extended training. The extended results confirm that while our approach underperforms in low-dimensional spaces, it achieves substantial gains as the embedding dimension increases, likely because higher-dimensional representations better capture hierarchical structures in the latent space.

Table 4: Comparison of HITS@10 and AUC-ROC scores for *Kinships* and *WN18RR* datasets averaged over 3 runs. The standard error of the mean for all cases is approximately 0.005.

| Model | Metric (D) | *Kinships* 10 | 20 | 30 | *WN18RR* 10 | 20 | 30 |
|---|---|---|---|---|---|---|---|
| RESCAL | HITS@10 | 0.924 | **0.975** | **0.970** | 0.029 | 0.049 | 0.060 |
| | AUC-ROC | 0.959 | **0.977** | **0.974** | 0.616 | 0.731 | 0.752 |
| BERNOULLI | HITS@10 | **0.964** | 0.930 | 0.870 | 0.362 | **0.415** | **0.424** |
| | AUC-ROC | **0.971** | 0.956 | 0.946 | 0.955 | 0.895 | 0.872 |
| BERNOULLI-RE | HITS@10 | 0.939 | 0.900 | 0.855 | 0.323 | 0.395 | 0.416 |
| | AUC-ROC | 0.958 | 0.945 | 0.935 | **0.963** | **0.957** | **0.952** |
| POISSON | HITS@10 | 0.941 | 0.921 | 0.862 | 0.325 | 0.397 | 0.418 |
| | AUC-ROC | 0.958 | 0.947 | 0.932 | 0.932 | 0.901 | 0.893 |
| LDM | HITS@10 | 0.948 | 0.950 | 0.943 | **0.365** | 0.346 | 0.356 |
| | AUC-ROC | 0.948 | 0.961 | 0.958 | 0.847 | 0.911 | 0.843 |
| HMSKGE | HITS@10 | 0.667 | 0.678 | 0.647 | 0.028 | 0.111 | 0.112 |
| | AUC-ROC | 0.916 | 0.913 | 0.910 | 0.865 | 0.917 | 0.917 |

Beyond predictive performance, we also evaluated the calibration of the predicted probabilities. The corresponding results, reported in Appendix C, suggest that HMSKGE produces well-calibrated probability estimates, as measured by the Brier Score.

Table 5: Comparison of HITS@10 and AUC-ROC scores for *FB15k-237* and *YAGO3-10* datasets averaged over 3 runs. The standard error of the mean for all cases is approximately 0.005.

| | Metric | *FB15k-237* | | | *YAGO3-10* | | |
|---|---|---|---|---|---|---|---|
| Model | (D) | 10 | 20 | 30 | 10 | 20 | 30 |
| RESCAL | HITS@10 | 0.208 | 0.335 | 0.366 | **0.105** | 0.122 | 0.143 |
| | AUC-ROC | 0.841 | 0.898 | 0.893 | 0.942 | 0.946 | 0.953 |
| BERNOULLI | HITS@10 | 0.438 | 0.438 | 0.407 | 0.067 | 0.065 | 0.072 |
| | AUC-ROC | **0.999** | 0.996 | 0.990 | **0.997** | 0.993 | 0.993 |
| BERNOULLI-RE | HITS@10 | **0.442** | **0.439** | 0.407 | 0.067 | 0.075 | 0.092 |
| | AUC-ROC | 0.942 | 0.877 | 0.859 | **0.997** | 0.996 | 0.995 |
| POISSON | HITS@10 | 0.439 | 0.445 | **0.439** | 0.090 | **0.220** | **0.248** |
| | AUC-ROC | **0.999** | **0.998** | **0.996** | **0.997** | **0.998** | **0.998** |
| LDM | HITS@10 | 0.407 | 0.430 | 0.434 | 0.057 | 0.070 | 0.114 |
| | AUC-ROC | 0.996 | 0.991 | 0.986 | 0.996 | 0.996 | 0.996 |
| HMSKGE | HITS@10 | 0.227 | 0.254 | 0.276 | 0.032 | 0.034 | 0.042 |
| | AUC-ROC | 0.995 | 0.995 | 0.943 | 0.938 | 0.929 | 0.935 |

In summary, the results presented in Tables 2, 3, 4, and 5 indicate that HMSKGE effectively captures relational patterns in knowledge graphs. However, its performance remains sensitive to the embedding dimensionality, with higher-dimensional embeddings enabling the model to more effectively represent the hierarchical structures employed in the approximation of the full likelihood. The experiments further demonstrate that gradient-based likelihood optimization offers a clear advantage over traditional alternating least squares methods, while the incorporation of random effects can additionally improve predictive performance. The differing behavior observed between HITS@10 and AUC-ROC suggests that the hierarchical approximation is particularly effective at capturing global discriminative structure, although this does not always translate into optimal top-rank predictions. Overall, the findings highlight the trade-offs among model capacity, embedding dimensionality, and evaluation criteria in achieving accurate link prediction across diverse knowledge graphs.

## 4.2 Taxonomy Induction

As described in Pietrasik & Reformat (2020), information in knowledge graphs is typically structured using an ontology, which provides semantics to the knowledge graph's triplets through an axiomatic foundation that defines the associations between entities and relations. A crucial component of most ontologies is the class taxonomy, organized through class subsumption axioms that represent is-a relationships between classes. This taxonomy often resembles a rooted tree, with a root class from which all other classes logically descend. The challenge of class taxonomy induction from knowledge graphs involves generating subsumption axioms from triplets to construct the class taxonomy.

To evaluate the clustering performance of the model, we applied the HMSKGE model with an embedding dimension of 64 to the *Kinships* knowledge graph. The resulting inferred taxonomy is shown in Figure 2. The *Kinships* graph represents a tribe in Australia, organized into four distinct communities. It contains metadata for each individual and defines 26 different types of relationships among tribe members. The inferred taxonomy exhibits a clear hierarchical organization: the first level clusters individuals by tribe affiliation, the second by sex, and the third by age. To quantitatively assess the alignment with the ground-truth hierarchy, we computed the Adjusted Rand Index (ARI) Hubert & Arabie (1985) at each level (Section, Sex, Age), obtaining scores of 1.00, 0.85, and 0.42, respectively. This example demonstrates that the hierarchical structure learned by the HMSKGE model effectively captures and organizes the underlying relational patterns in the knowledge graph.

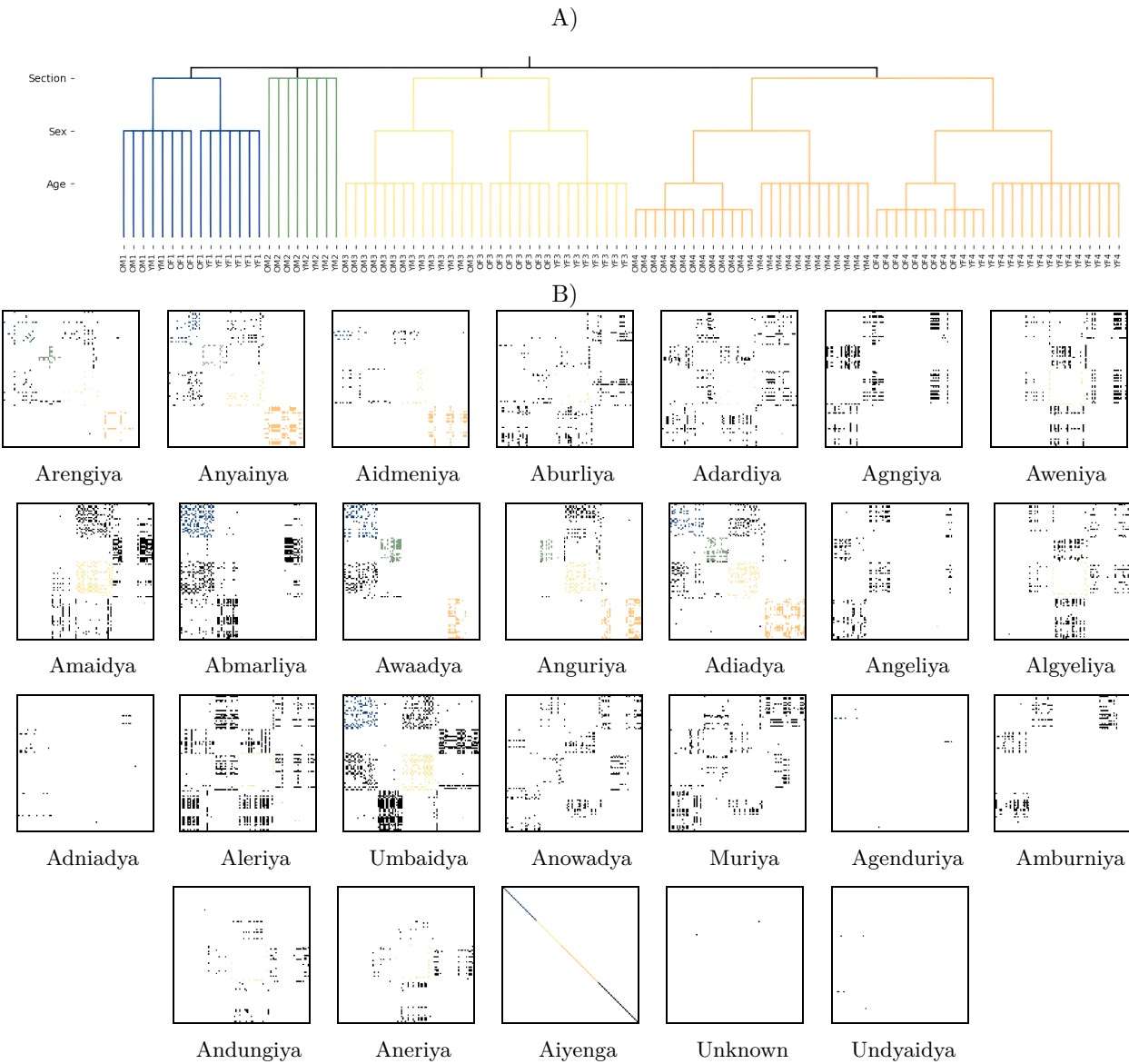

Figure 2: **A)** The dendrogram illustrates the *Kinships* dataset, where 'O' denotes old age, 'Y' denotes young age, 'M' denotes male, 'F' denotes female, and numbers represent different families within the tribe. **B)** The adjacency matrices per relation in the *Kinships* knowledge graph, reordered according to the inferred hierarchy given at the top.

We additionally visualize the smallest *WN18RR* of the larger knowledge graph providing a complementary perspective on the clustering behavior of the HMSKGE model for the visualization configured with an embedding dimension of 16. As shown in Figure 3, the adjacency matrices corresponding to different relation types exhibit clear block structures, indicating that nodes sharing similar semantic or relational roles are grouped together. Some faint lines can also be observed across several matrices; these appear to result from the random effects inherent to our model, which in turn enable it to capture hub nodes accurately.

Overall, these results confirm that HMSKGE effectively captures hierarchical and relational structures in knowledge graphs. By leveraging both entity interactions and relation-specific patterns, the model produces interpretable clusters and taxonomies that reflect meaningful latent groupings. Visualizations of adjacency matrices and dendrograms further illustrate how entities are organized according to their relational roles,

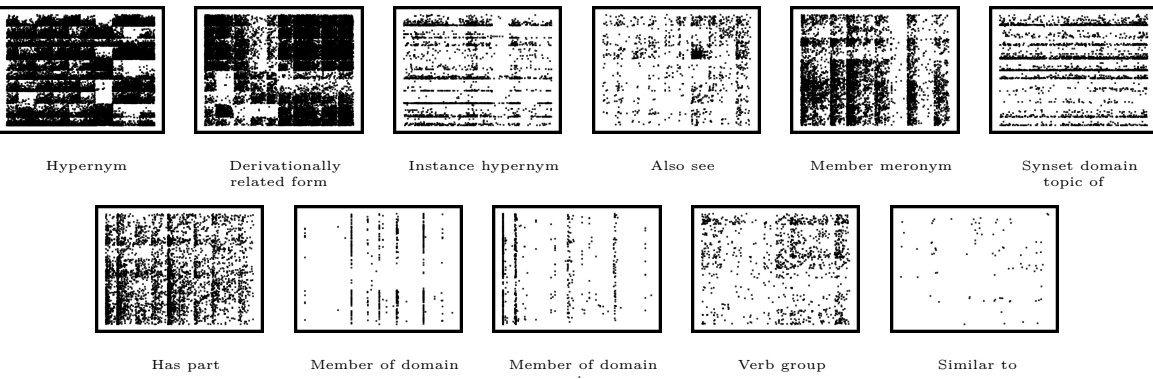

Figure 3: Adjacency matrices for each relation in the *WN18RR* knowledge graph, reordered according to clusters inferred by HMSKGE. Each matrix shows the connectivity pattern for a specific relation, with entities arranged to highlight relationally coherent clusters. Clear block structures indicate that entities sharing similar relational roles are grouped together.

providing clear insights into the structure of the graphs. While these results are dataset-specific, they highlight the HMSKGE model's ability to uncover interpretable hierarchies and clusters in knowledge graphs.

## 5   Conclusions

The proposed methodology presents a novel approach to knowledge graph embeddings by incorporating latent space distances and hierarchical clustering directly into the training process. This integration brings multiple advantages: first, the method is computationally efficient, scaling quadratically with the embedding dimensionality, which boosts scalability for larger graphs. Second, approximating the full likelihood eliminates the need for negative sampling, enabling it to capture the complete structure of the knowledge graph. Moreover, the embeddings effectively preserve hierarchical relationships, forming an ontology that mirrors the graph's inherent structures. Applied across four diverse knowledge graphs, this approach achieves competitive link prediction performance when the model used higher dimensional embeddings (i.e. $D \geq 10$) thus providing sufficient capacity for the hierarchical approximation to characterize the relational structure, whereas we observed the model to underperform in the low capacity regime (i.e., $D = 2$ and $D = 3$). Notably, the inferred hierarchy can be used as a data-driven approach to learn taxonomic representation in knowledge systems thereby naturally bridging knowledge graph embedding approaches with concept hierarchy learning.

### 5.1   Limitations and broader impact

The inference procedures are subject to local minima and are not guaranteed to identify the optimal representation of knowledge graphs. Furthermore, the learning of hierarchies by use of clustering is NP-hard and prone to local minima issues. Whereas the full-likelihood approximation circumvents the need for negative sampling it still relies on the accuracy of the approximation which especially for low-dimensional embeddings can be coarse. Care has to be taken when using the model structure to infer missing or unobserved relations as the link prediction in some cases is far from perfect and may result in wrong conclusions in regard to entity-relationships. Caution is also warranted when interpreting entity relations, as the learned representations may inherit and reproduce biases present in the underlying data. Practitioners should prefer HMSKGE when the primary objective is structural inference and the exploration of hierarchical organization, rather than optimization of ranking performance. In particular, HMSKGE is well-suited for studying the intrinsic structure of a graph, where emphasis is placed on organizing nodes according to their relational patterns rather than relying on node metadata. This approach is especially appropriate when the goal is to determine whether latent hierarchical structures exist within the graph and, if so, to characterize the nature of these hierarchies. However, the approach can be used to efficiently probe facts unobserved to be verified or dismissed. Whereas we considered the widely used and well-established RESCAL framework, the

approach readily extends to existing distance-based knowledge graph embedding procedures. Future work should investigate if other knowledge representation modeling procedures than RESCAL can be reformulated in terms of distance models amenable to similar scalable hierarchical approximations as the proposed HMSKGE.

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

## A  Learning Curves

In Figure 4, the learning curves for all models applied to the *Kinships* and *WN18RR* knowledge graphs are presented. The results show that all models converge to a similar loss, indicating that *HMSKGE* effectively approximates the performance of the comparison models. We also observe that, in both cases, convergence is faster and smoother when using larger embedding dimensions.

## B  Clustering

To assess the model's behavior in a controlled environment, HMSKGE was applied to an artificial knowledge graph comprising 30 entities and 3 relations, arranged into three predefined clusters. Each cluster interacted internally and with other clusters through specific relations. After randomizing the node order, the model successfully recovered the underlying clusters, as illustrated in Figure 5, achieving an ARI of 1. This experiment demonstrates that HMSKGE can robustly infer cluster structure from relational patterns even when entity ordering provides no prior cues.

## C  Calibration

To complement the link prediction results, we assess the calibration of the models on the *Kinships* dataset using the Brier Score, defined as the mean squared error between predicted probabilities and true binary labels. Lower values indicate better-calibrated probability estimates. All models were trained for 20,000 epochs with a learning rate of 0.01. Table 6 reports results across all embedding dimensions considered.

HMSKGE achieves the best calibration across all embedding dimensions. The BERNOULLI and BERNOULLI-RE models level off at a Brier Score of about 0.525, indicating increasingly uniform probability estimates. In contrast, HMSKGE maintains low Brier Scores across all dimensions, suggesting that it produces more reliable and well-calibrated probability estimates.

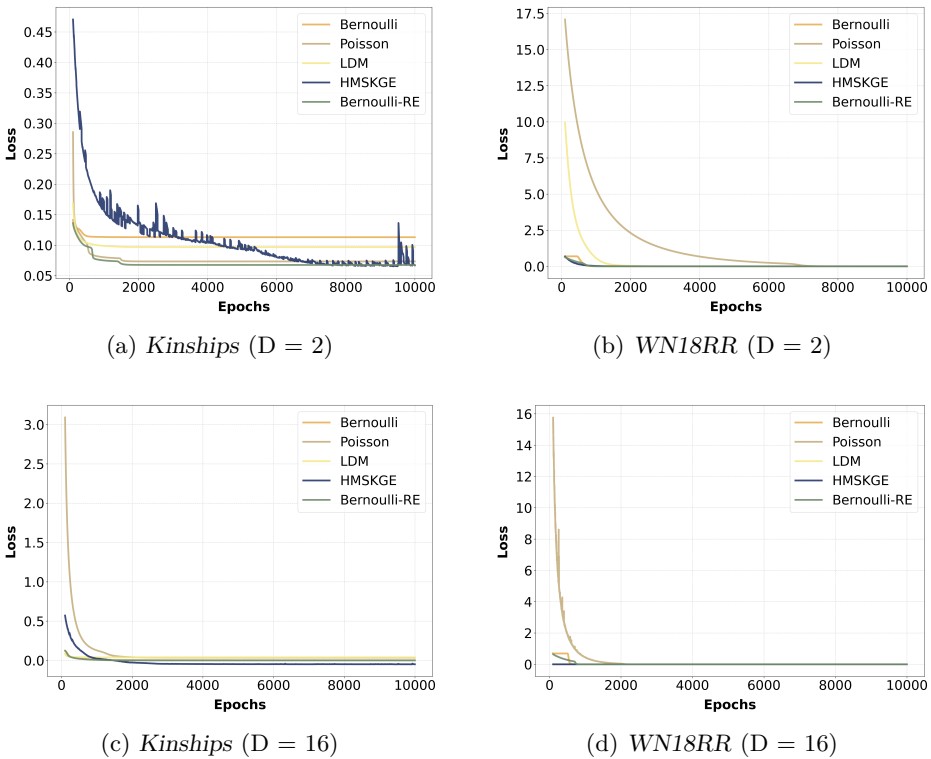

(a) *Kinships* (D = 2)                    (b) *WN18RR* (D = 2)

(c) *Kinships* (D = 16)                   (d) *WN18RR* (D = 16)

Figure 4: Learning curves for *Kinships* and *WN18RR* for dimensions 2 and 16. Plots start from epoch 100. HMSKGE losses are computed using the hierarchical approximation, whereas other models use the batched likelihood.

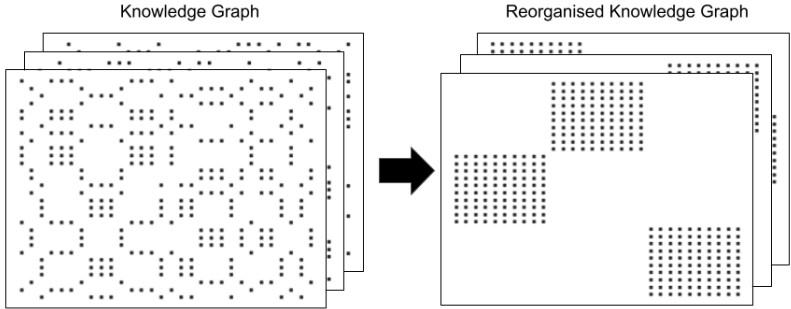

Figure 5: Reorganization of an artificial knowledge graph using HMSKGE, showing accurate recovery of three predefined clusters from randomized node order.

Table 6: Brier Score for models on the *Kinships* dataset across all embedding dimensions.

| | *Kinships* | | | | | |
| Model | $D = 2$ | $D = 3$ | $D = 10$ | $D = 16$ | $D = 20$ | $D = 30$ |
|---|---|---|---|---|---|---|
| BERNOULLI | 0.442 | 0.405 | 0.198 | 0.464 | 0.525 | 0.525 |
| BERNOULLI-RE | 0.291 | 0.231 | 0.311 | 0.525 | 0.525 | 0.525 |
| POISSON | 0.342 | 0.293 | 0.304 | 0.396 | 0.479 | 0.515 |
| LDM | 0.349 | 0.295 | 0.282 | 0.322 | 0.355 | 0.415 |
| HMSKGE | **0.221** | **0.169** | **0.112** | **0.115** | **0.127** | **0.129** |

