# OpenReview forum: "Hierarchically Metric-Structured Knowledge Graph Embeddings"
_TMLR — Under review for TMLR_

### Review · Reviewer_RDcP · 2026-01-03

**Summary Of Contributions:**

This paper proposes Hierarchically Metric-Structured Knowledge Graph Embeddings (HMSKGE), a knowledge graph embedding model that combines a RESCAL-style latent factorization with a distance-based latent space and a hierarchical (dendrogram-based) approximation to the full likelihood. The main goal is to avoid negative sampling while enabling scalable inference and providing interpretable multiscale structure over entities.

The paper’s key strengths are a clear probabilistic motivation, a well-structured modeling narrative, and the attempt to unify link prediction with hierarchy discovery. However, empirically the method underperforms strong likelihood-based baselines on standard ranking metrics (e.g., HITS@10), and the benefits of the hierarchical structure are not fully demonstrated quantitatively.

**Additional Comments:**

Overall, this is a thoughtful and technically interesting paper, but its current positioning overstates empirical performance relative to the presented results. With clearer framing, stronger evaluation of the hierarchical component, and better integration with recent hierarchy-aware methods such as KG-FIT (NeurIPS 2024), the paper would be substantially improved and more compelling to the TMLR audience.

**Audience:**

Yes

**Audience Explanation:**

Researchers interested in probabilistic KGE models, alternatives to negative sampling, and interpretable or hierarchical structure in knowledge graphs would find this work relevant. The connection between latent distance models, likelihood approximation, and hierarchy induction is conceptually interesting and aligns well with ongoing work on structure-aware and hierarchy-aware KG embeddings.

**Broader Impact Concerns:**

No major concerns. As with most KG embedding methods, the learned representations may reflect biases present in the underlying data. A brief acknowledgment of this would be sufficient.

**Claims And Evidence:**

No

**Claims Explanation:**

While the modeling framework is technically sound and clearly described, the experimental results do not fully support the paper’s claims of strong predictive performance. Across multiple benchmarks, HMSKGE shows substantially lower HITS@10 compared to competing likelihood-based models, even when AUC-ROC is competitive. The paper does not clearly explain how these metric discrepancies should be interpreted in practice.

If the main contribution is interpretability and scalable approximation rather than link prediction accuracy, this should be stated more explicitly and supported with stronger empirical evidence focused on hierarchy quality or scalability.

**Requested Changes:**

**Critical**

1. Clarify evaluation metrics. The paper gives mixed messages about the role of AUC-ROC versus ranking-based metrics like HITS@10. The authors should clearly explain which metric is most appropriate for their setting and how to interpret cases where they disagree.
2. Strengthen empirical support for the main contribution. Either:
    * provide quantitative evaluation of the learned hierarchies,
    * demonstrate clear scalability or computational advantages, or
    * explicitly reframe the contribution away from competitive link prediction performance.
3. Discuss and cite KG-FIT and related hierarchy-based KGE methods. Recent work such as KG-FIT, which constructs and exploits entity hierarchies to improve KG embeddings, is highly relevant and should be cited. A short comparison clarifying differences in how hierarchies are constructed and used would significantly strengthen the related work section.

**Non-critical**

* Consider reporting additional standard KGE metrics (e.g., MRR, HITS@1), or justify their omission.
* More clearly articulate when a practitioner should prefer HMSKGE despite weaker ranking performance.

---

> ### Author Response · Authors · 2026-06-13
>
> We thank the reviewer for the thoughtful and constructive comments. We have revised the manuscript according to the reviewer's suggestions and addressed the reviewer's concerns below. All changes in the revised manuscript have been indicated by red text.
>
> Re. Claims of strong predictive performance
>
> We have revised the manuscript to better align the claims with the empirical evidence and strengthened the evaluation of the hierarchical component (see third bullet in the Introduction and Section 3.2). We agree that the original version unintentionally overstated link prediction performance in the conclusion section, and we have accordingly revised the conclusion. We also note that concerns regarding scalability and computational efficiency are addressed in Section 3.2.2 through the discussion of the computational complexity and scaling behavior of HMSKGE. See also our response to reviewer JCfH.
>
> Re. Clarify evaluation metrics.
>
> To address the mixed behavior of evaluation metrics, we added a clearer discussion explaining the roles of AUC-ROC and ranking-based metrics. In particular, the revised Section 4.1 now explains that metrics such as HITS@10 and AUC-ROC capture complementary aspects of link prediction performance, with AUC-ROC assessing the model’s global discriminative ability between true and false triples and HITS@10 evaluating the extent to which correct entities are prioritized among the top-ranked predictions. We focus on HITS@10 and AUC-ROC because our objective is to evaluate whether HMSKGE retains useful ranking and discrimination ability while introducing a hierarchical structure, rather than to provide an exhaustive benchmark of ranking metrics. We further discuss in the revised manuscript that the HMSKGE procedure generally performs more favorably on the AUC-ROC metric than the HITS@10 metric, which we attribute to the hierarchical approximation well accounting for the global separability.
>
> Re. Strengthen empirical support for the main contribution.
>
> To strengthen empirical support for the hierarchical aspect, we added quantitative clustering and block-structure evaluation measures to complement the previous visual analysis. For the kinship data, where ground-truth labels are available, and for the artificial knowledge graph, we evaluated the inferred clusters using the Adjusted Rand Index (ARI) (see the revised Section 4.2 and Appendix B). The obtained ARI scores demonstrate that the inferred hierarchical organization is well aligned with the underlying relational structure and ground-truth grouping patterns. Additionally, we report Brier Score results for the Kinships dataset across all embedding dimensions in a new Appendix C. HMSKGE achieves the lowest Brier Score across all settings, suggesting that the hierarchical full-likelihood approximation may yield well-calibrated probability estimates. We further clarify in the revised manuscript’s introduction and new related work section that our approach naturally bridges conventional knowledge graph embedding approaches with hierarchical representation learning of concepts within a unified probabilistic framework by invoking a multi-scale hierarchical clustering structure designed to optimally characterize the knowledge graph by approximating the full data likelihood.
>
> Re. Discuss and cite KG-FIT and related hierarchy-based KGE methods.
>
> We created a related work section and included recent hierarchy-aware KGE methods, including KG-FIT, in the new related works section (new section 2 in the manuscript).
>
> Re. Broader impact
>
> We further added a clearer discussion of when a practitioner should prefer HMSKGE, namely in settings where interpretability, multiscale structure, and scalable likelihood-based inference are more important than solely link prediction peak ranking performance (see Section 5.1).

---

### Review · Reviewer_JCfH · 2026-01-08

**Summary Of Contributions:**

The paper proposes a way to embed knowledge graphs under the assumption that they have some hierarchical structure that improves the embedding w.r.t. tasks like link prediction and taxonomy induction.
For the construction of the dendrogram, k-means is used and the authors compare their new method HMSKGE with some baseline methods on four datasets/ knowledge graphs.
The overall idea is reasonable, however, there are some issues that need to be solved (see below) and even then, the comparison with baseline methods is not convincing.

**Audience:**

Yes

**Audience Explanation:**

The topic is relevant to TMLR and including hierarchical structure of knowledge graphs into the embedding is an idea worth pursuing.

**Broader Impact Concerns:**

No concerns.

**Claims And Evidence:**

No

**Claims Explanation:**

W1) The results in the experiment section show that HMSKGE is usually worse than state-of-the-art methods. Especially the conclusion is overselling the contributions of the paper in stating that HMSKGE "achieves competitive link prediction performance against numerous baseline models, even when using low-dimensional embeddings". In fact, all tables indicate the opposite. E.g., HMSKGE performs *clearly worst among all methods* for dataset FBK15K-237 for D=2 and D=3 and on YAGO3-10 for D=2 and Kinships for D=3 and D=16 and WN18RR for D=3. I can only find one case where HMSKGE is better than its competitors (WN18RR with D=16). On page 8, the authors themselves state that "our approach underperforms in low-dimensional spaces".

W2) The related work section is missing. While the authors introduce the general area of knowledge graphs in the introduction, I'm missing more explanations and comparisons with state-of-the-art methods. Notably, I see a strong connection to deep clustering methods that also include (potentially hierarchical) clustering structure into the embedding by adapting the loss function. E.g., [1] can serve as a starting point for the literature research.

W3) Some formulas should be accompanied by more explanations, especially Equation 6.

W4) Page 11: The block structure is only analyzed visually where there are measures to evaluate objectively, see, e.g., [2]

W5) Large parts of figure 1 are not readable.

[1] Guo, X., Gao, L., Liu, X., & Yin, J. (2017, August). Improved deep embedded clustering with local structure preservation. In Ijcai (Vol. 17, pp. 1753-1759).

[2] Chakrabarti, D., Papadimitriou, S., Modha, D. S., & Faloutsos, C. (2004, August). Fully automatic cross-associations. In Proceedings of the tenth ACM SIGKDD international conference on Knowledge discovery and data mining (pp. 79-88).

**Requested Changes:**

Please tackle above weak points, especially:

C1) I think the method needs some adaptions in order to work better for the targeted tasks. E.g., using iterative k-Means when looking for a hierarchical structure is suboptimal. Why not use Single Linkage agglomerative clustering, for example? That could be tested.
Also, add explanations when and why your method should be used-- are there KGs known to have a hierarchical structure? Perform synthetic experiments for that. Which influence does the strength of hierarchical structure have on the results?

C2) Please add a related work section and investigate in how far the above mentioned related work concerns your method or can help to improve it.

W3-W5.

Minor:

M1) Wrong authors in references on page 15, "You can Teach" is not an author.

M2) Use $\varepsilon$ (varepsilon) instead of $\epsilon$ (epsilon)

---

> ### Author Response · Authors · 2026-06-13
>
> We thank the reviewer for the thoughtful and constructive feedback. We have revised the manuscript accordingly and addressed the reviewer’s concerns below. All changes in the revised manuscript have been indicated in red.
>
> Re. W1)
>
> We agree that the paper unintentionally overstated that the statement in the conclusion was inaccurate. What we meant to state was that even in the low-dimensional case, providing a low capacity regime for the HMSKGE, we observed reasonable performance. However, we agree that the original conclusions overstated the empirical performance, which was not our intention, and as also stated in our discussion, the HMSKGE performance in the low-dimensional case is indeed hampered by the hierarchical approximation. We have therefore carefully revised the manuscript and, in particular, the conclusion to make this clear and to ensure results are not overstated. Specifically, we have clarified in the revised conclusion that the HMSKGE does not outperform state-of-the-art methods in general, and further elaborated upon the hampered performance in the low-dimensional settings where the hierarchical approximation in this low-capacity regime induces a coarse approximation. Additionally, we report Brier Score results for the Kinships dataset across all embedding dimensions in a new Appendix C. On this dataset, HMSKGE achieves the lowest Brier Score across all considered settings, indicating that the hierarchical full-likelihood approximation may provide well-calibrated probability estimates in this particular case.
>
> Re. C1)
>
> Thanks for raising this point. We disagree that iterative k-means is suboptimal, and importantly, this strategy was specifically chosen due to its scaling when compared to agglomerative approaches. In contrast to agglomerative methods such as single-linkage clustering, it does not require computing all pairwise distances, which becomes impractical for large knowledge graphs. This makes k-means a more suitable choice for our setting. Importantly, the k-means objective directly minimizes the squared Euclidean norm, which our distance-based formulation explicitly relies upon. Whereas single linkage agglomerative clustering can proxy the k-means objective, the single linkage agglomeration will emphasize nearby points between clusters as opposed to the overall inter-cluster distances between the observations of the merged clusters. Consequently, average linkage and, in particular, Ward’s method would more directly reflect an average estimate of the squared Euclidean distance similar to our employed K-means procedure. However, K-means scales linearly in the number of observations, whereas these agglomerative hierarchical clustering procedures rely on all pairwise distances, thus producing less favorable scaling properties. We have in the Methods section 3.2.2, when discussing the complexity of the HMSKGE, we elaborated on our choice using the k-means clustering procedure as opposed to agglomerative hierarchical clustering approaches. We additionally expanded the discussion in the limitations section to clarify when HMSKGE is most appropriate for practitioners to use.
>
> Re. W2 and C2)
>
> We thank the reviewer for this suggestion, and we have added a related work section (new Section 2) in the revised manuscript, discussing prior work on knowledge graphs and clarifying how our approach relates to these existing methods. We also considered the suggested deep clustering literature. However, we did not elaborate further on these methods, as the primary focus of our work is on knowledge graph-based approaches.
>
> Re. W3)
>
> We thank the reviewer for pointing this out. We have included additional explanations to clarify the relevant equations (see revised Section 3.2).
>
> Re. W4)
>
> We complemented the block structure analysis in the revised manuscript with quantitative evaluation measures. Specifically, for the kinship knowledge graph, where ground-truth labels are available, and for the artificial knowledge graph, the inferred clusters were evaluated using the Adjusted Rand Index (ARI). The corresponding results and discussion are presented in Section 3.2 and Appendix B.
>
> Re. W5)
>
> Thanks for pointing this out. We have revised Figure 1 to make it less cluttered. In the text, we have removed some of the equations, leaving only the central model structures imposed for the figure. We have further improved the resolution of the figure.
>
> All minor issues raised by the reviewer were addressed, including reference errors and notation.